# The Effect of Caffeinated Chewing Gum on Volleyball-Specific Skills and Physical Performance in Volleyball Players

**DOI:** 10.3390/nu15010091

**Published:** 2022-12-24

**Authors:** Magdalena Kaszuba, Olga Klocek, Michał Spieszny, Aleksandra Filip-Stachnik

**Affiliations:** 1Institute of Sport Sciences, The Jerzy Kukuczka Academy of Physical Education in Katowice, 40-065 Katowice, Poland; 2Institute of Sport Sciences, University of Physical Education in Krakow, 31-571 Krakow, Poland

**Keywords:** ergogenic aid, team sports, sports nutrition, supplement

## Abstract

No previous study analyzed the effect of caffeinated chewing gum on volleyball-specific skills and physical performance in volleyball players. Twelve volleyball players participated ina randomized, crossover, and double-blind experiment after ingestion of (a) ~3.2 ± 0.4 mg/kg of body mass (BM) of caffeine via chewing gum or (b) non-caffeinated chewing gum (placebo) and performed: (a) a countermovement jump, (b) a squat jump, (c), an attack jump, (d) a block jump, (e) 5 and 10 m sprints, (f) a modified agility *t*-test, (g) an attack and service speed test, and (h) a spike and serve accuracy test. Compared to the placebo, the caffeine chewing gum supplementation significantly improved the accuracy of the attack (15 ± 4 vs. 18 ± 3 points, *p* = 0.02). However, the ingestion of caffeinated chewing gum had no effect on the remaining other performance tests (*p* from 0.12 to 1.00). A caffeine-containing chewing gum with a dose of ~3 mg/kg BM effectively improved the attack’s accuracy in volleyball players. However, this effect was not observed in better results in jumping, running, and other skill-based volleyball tests.

## 1. Introduction

Volleyball is a physically demanding team sport with technical and tactical requirements [1]. This sport discipline is predominately based on anaerobic pathways [2] because players perform numerous high-intensity efforts with short recovery periods (~3–9 s), including acceleration, deceleration, jumping, frequent changes of directions, and actions with the ball [3,4]. In addition, volleyball has many requirements that combine physical and technical skills that determine the majority of the game’s elements, including attacking, serving, blocking, and setting [5].

Caffeine is considered the most popular psychoactive drug used by the majority of the general population in the world [6]. Due to its ergogenic effect, it is widely used by athletes of many sports disciplines. Interestingly, between 1984 and 2004 caffeine was banned in sports competitions, although only in extremely high doses (i.e., representing a concentration of caffeine greater than 12 μg/mL in urine) [7]. However, on 1 January 2004, the World Anti-Doping Agency (WADA) decided to remove caffeine from the list of banned substances. Since then, athletes can freely use this substance during competitions. Nevertheless, caffeine has been moved to the Monitoring Program to control the consumption of high doses of caffeine by athletes (i.e., over 6 μg/mL in urine) as they can be harmful to them [8]. Interestingly, a urinary caffeine concentration greater than 15 µg/mLis still prohibited by the National Collegiate Athletic Association (NCAA) [9]. However, since the ban was lifted for almost all associations, caffeine-containing supplements have been hugely popular in the sports world. It has been shown that up to 75–90% of athletes use caffeine before or during competition [6]. The recommended doses of caffeine intake by athletes are 3–6 mg/kg body mass (BM). This is due to the fact that higher doses of caffeine (9–13 mg/kg BM) do not result in additional benefits, and also increase the risk of side effects [10,11].

Previous studies have shown that caffeine has a positive effect on various types of exercise, such as intensive efforts of short duration [12], resistance exercises [13], and performance in team sports [14]. So far, it has been proven that caffeine improves countermovement and squat jump performance [15,16,17], as well as volleyball-specific jumping techniques and agility [15,16] in volleyball players. Moreover, caffeine positively influenced typical volleyball activities, such as the ball’s spikes velocity of the ball and increased the number of successful actions during the real match conditions [15,16]. However, in the studies mentioned above, the effect of caffeine was evaluated after a dose of 3 mg/kg BM supplemented in the form of an energy drink [15,16] or a capsule at a dose of 5 mg/kg BM [17]. It is worth noting that caffeine is also available in gels, bars and chewing gums [18]. However, less attention has been paid to those caffeine sources in previous research focused on volleyball performance.

In most of the studies and real sports scenarios, caffeine is typically consumed 45–60 min before exercise in the form of capsules, coffee, or an energy drink [18]. However, it can be problematic when fast caffeine absorption is desired. Interestingly, caffeinated chewing gum offers a faster method of delivery; after 5 min of chewing, even 85% of this supplement is bioavailable [19]. Due to the quick absorption, this route of administration might be a valuable alternative for volleyball players to use just before or during the match. Unfortunately, studies evaluating the effectiveness of caffeine supplementation via chewing gum are scarce, and those available provide inconclusive findings. For example, Paton et al. [20] showed a significant decline in the power output rate between sets in a cycling test after intake of ~3 mg/kg BM caffeinated gum in elite cyclists. Similarly, the study by Filip-Stachnik et al. [21] did not find a positive effect of ~2.7 mg/kg and ~5.4 mg/kg BM of caffeine via chewing gum on repeated special judo fitness tests. In terms of using this form of caffeine in team sports, improvement in attack jump height has been found after ~6 mg/kg BM of caffeine among volleyball players [22]. However, this study only assessed the effects of caffeine on jumping performance and the actions during a simulated game. Thus impact on specific skill-based volleyball performance is still unknown.

Therefore, the aim of this study was to determine the effect of caffeinated chewing gum in a dose of ~3 mg/kg BM on volleyball-specific skills and physical performance in volleyball players. To the best of our knowledge, this is the first study to evaluate the effects of caffeinated chewing gum on both numerous performance tests and specialized volleyball skills tests.

## 2. Materials and Methods

### 2.1. Participants

The study included 12 healthy volleyball players, of which nine are men and three are women (age = 23 ± 3 years; body mass = 85.9 ± 11.2 kg; height = 188 ± 8 cm; body mass index (BMI) = 24.4 ± 1.7; habitual caffeine intake = 2.7 ± 2.2 mg/kg/day) with at least 5 years of volleyball training experience (9 ± 4 years). The inclusion criteria were as follows: (a) free from neuromuscular and musculoskeletal disorders; (b) minimum 5 years of experience in volleyball training; (c) no medication nor dietary supplements used within the previous month which could potentially affect the study outcomes (e.g., beta-alanine, creatine, etc.); (d) self-described good health status; (e) the same phase of the menstrual cycle in both trials (this criterion was provided because previous studies showed that in different phases of the menstrual cycle, women are liable to hormonal changes that can affect sports performance [10] and the rate of caffeine metabolism [7]); and (f) no use of hormonal contraception. Players were excluded if they reported: (a) positive smoking status (since the metabolism of caffeine is doubly accelerated in cigarette smokers [7,23,24]); and (b) a potential allergy to caffeine. Players were recruited from the same volleyball team and testing was made at once after the competitive season. The Bioethics Committee approved the study protocol at the Academy of Physical Education in Katowice, Poland (3/2019), per the latest version of the Declaration of Helsinki. All players provided their written informed consent before participating in this study. The sample size was determined using G*Power version 3.1.9.2 (Dusseldorf, Germany), and the following parameters “*t*-test, difference between two dependent means” were assumed as a statistical test (1 group of subjects, 2 experimental conditions); the statistical power was 0.8, the significance level was 0.05, and the effect size was 1.22 based on previous study [25] that investigated the acute effect of caffeine on physical performance of volleyball players. The power analysis indicated that a minimum sample size of 8 individuals was required for this study.

### 2.2. Pre-Experimental Standardization

Before the first experimental trial, study players were advised to maintain their standard dietary and hydration habits throughout the study and to maintain their habitual caffeine intake. In order to achieve objective dietary standardization, players repeated the same dietary regimen before each trial. Moreover, the collected data was verified by a qualified dietitian. The menstrual cycle was determined using a period tracker application. The regularity and length of each participant’s menstrual cycle were monitored for 4 months prior to the investigation using a mobile application (Mycalendar-Period Tracker, Louisville, KY, USA) [26]. On this basis, it was determined in which phase of the menstrual cycle the players were during the study. Habitual consumption of caffeine was determined usinga modified version of the validated questionnaire by Bühler et al. [27], which specified the type and amount of caffeine-containing products. Habitual caffeine intake was assessed for 4 weeks prior to the experiment, as previously recommended. Based on the proposed classification [28] thresholds for habitual caffeine intake, players were classified as mild-moderate caffeine users because they ingested on average 2.7 ± 2.2 mg of caffeine per day. Players were instructed to refrain from any sources of caffeine and alcohol 24 h before each experimental study, and not to engage in strenuous exercise for 24 h before testing.

### 2.3. Experimental Design

A randomized, crossover, double-blind design was used to evaluate the effects of the caffeinated chewing gum. Randomization was performed by a research team member who was not involved in the data collection. Therefore, when assigned to intervene, participants and investigators were blinded on each trial. Each player took part in one familiarization session and two identical experimental sessions with an interval of 72 h between trials to ensure complete recovery [29,30] and substance wash-out. The players arrived at the volleyball court at the same time of the day at 6 p.m., which was their habitual training time. The experimental sessions included the ingestion of caffeinated chewing gum 15 min before the start of the tests as follows: (a) the ingestion of caffeine-free chewing gum, acting as placebo (PLAC); (b) ~3.2 ± 0.4 mg/kg BM of caffeine (CAF) (range: 2.7–4.0 mg/kg BM) in the form of chewing gum. Caffeine in the form of chewing gum was served in an absolute dose of 300 mg for men and 200 mg for women (Energisant; One Gum; Paris, France), (representing a dose similar to their normal daily caffeine intake). An absolute dose of caffeine was used instead of a weight-adjusted dose as it was technically impossible to prepare an individual treatment. However, different absolute doses of caffeine in men and women provided a similar value of the final individual supplementation in both groups (3.3 ± 0.3 mg/kg BM of caffeine in men vs. 2.8 ± 0.1 mg/kg BM of caffeine in women). The placebo was a commercially available decaffeinated chewing gum of similar flavor, color, and shape (Airways, Warsaw, Poland). The gums were cut into small pieces and placed in an opaque cup to assist players and researchers in blinding. Players ingested chewing gum 15 min before each experimental trial, which is a common supplementation protocol for this form of caffeine [22]. Before starting the warm-up, players chewed gum for 5 min [19] and then spat it into a container. Next, the players performed a standardized, habitual pre-training warm-up, which lasted about 15 min. Then the players performed volleyball-specific skills and physical tests, which included the assessment of: (a) a countermovement jump (CMJ), (b) a squat jump (SJ), (c) an attack jump, (d) a block jump, (e) 5 and 10 m sprints, (f) a modified agility *t*-test, (g) an attack and service speed test, and (h)a spike and serve accuracy test. At the end of each trial, players were asked about possible side effects from caffeine ingestion and how they perceived improvement in performance during the study.

### 2.4. Jumping Assessment

After the warm-up, players performed the CMJ, SJ, attack jump, and block jump. Players performed 3 jumps in each trial, with 1 minute’s rest between them. The technique of making individual jumps was as follows. The CMJ was performed from a standing position with a straight torso, knees fully extended with hands on hips, and feet shoulder-width apart. Players descended to their usual squat depth and jumped for maximum height, and landed in an athletic position [31]. The SJ was performed with hands on hips, and the movement was started from a half-squat position, in which the players paused for 3 s to exclude the stretch and shorten the cycle, and then made a concentric movement [32]. The attack jump was performed with an approach of 2–3 steps leading to a jump with an arm swing. They were also instructed to jump as high as possible while maintaining the individual technique used, which is similar to their habitual jump but without the ball [2]. During the block jump, the players’ hands were placed at the level of the chest, and they made an upward jump consisting of the eccentric and concentric phases. The depth of the eccentric movement and the type of arm work followed the player’s match technique. After the jump, the arms were fully extended above the head to reach as high as possible [2]. The best jump of the three attempts for each jump type was used for further analysis. The jump height was measured with Optojump photoelectric cells (Microgate, Bolzano, Italy), which has been considered a valid and reliable tool for assessing vertical jump performance [33].

### 2.5. Running Assessment

The running time was assessed with 5 m and 10 m straight sprints, and a simple running agility test (modified *t*-test) was used to determine the speed with direction changes. The modified *t*-test (Figure 1) was performed based on the protocol proposed by Pauole et al. [34]. The modified *t*-test started with running forward to cone B, the base of which had to be touched with the right hand. Then the players facing the same direction moved left towards cone C, which had to be touched with the left hand. Next, the players shuffled right to cone D and touched it with their right hand. Then they went back to cone B andtouching it again. Finally, they ran backward to the starting line to complete the test. The test was repeated when the player did not touch the base of the cone with his/her hand, crossed his/her feet while shuffling sideways, or failed to face forward throughout. The running tests were measured by a set of photocells (Witty, Microgate, Bolzano, Italy), which was used to record the time of running tests in other authors’ research [35,36]. The tests were started from a standing position with feet placed 0.3 m before the starting line with the first photocells gate. The players were instructed to run as quickly as possible from a standing position, always starting with the same preferred leg [37]. The players made 2 trials of each test with 2 min rest between trials, and the best time of two attempts was used for statistical analysis. The accuracy of the measurement was 0.01 s.

### 2.6. Ball Velocity during Attack and Volleyball Service

The protocol of these exercises is based on a special testing protocol for monitoring spike and serve speed in volleyball published by Palao and Valades [38]. In the maximal standing spike test (Figure 2), the players were instructed to hit the ball as hard as possible by self-tossing without jumping. They were supposed to hit the designated zone (1.5 m × 1.5 m), which would be 4 m from the measuring point. The ball velocity of attack and volleyball service tests were performed on a standard volleyball court (9 m × 18 m) and at a constant net height of 2.24 m for women and 2.43 m for men. In the attack test (Figure 3), the players were instructed to hit the ball over the net with maximum force toward the designated zone linearly. The same setter set the ball to ensure standardization across each trial. The players used their natural attacking technique in their specific play zone during the attack. In the volleyball service test (Figure 4), the players hit the ball over the net with maximum strength towardsthe opponent’s court. They performed this test using their standard serving technique with a self-toss. In each test, the players made 3 attempts with a 30-s rest period between repetitions [38], and the best result of the three trials was used for further analysis. The ball velocity was measured with a radar gun (Velocity Speed Gun, Bushnell; Overland Park, KS, USA) that has been used in another publication [39]. A standard ball (Molten V5M5000) was used for tests, which players use during training and matches.

### 2.7. Volleyball Specific-Skills Assessment

Volleyball skills were assessed in terms of the accuracy of the attack and the service. These tests were performed on a standard volleyball court (9 m × 18 m) and at a constant net height of 2.24 m for women and 2.43 m for men. A modified test designed to standardize skill assessment for junior volleyball players was used to assess the accuracy of the attack [40]. The court has been divided into segments with a point value assigned to them (Figure 5). If the players hit the green zone (1 m × 1 m), they received 3 points; if they hit the yellow zone (2 m × 2 m), they got 2 points. The 1 point was awarded for hitting the ball in the court but outside the designated zones, while an error in the attack (ball in out or in the net) meant 0 points in a given attempt. The same setter set the ball to ensure standardization across each trial. The players used their natural attacking technique in their specific play zone during the attack. Instead, the accuracy of the volleyball service was assessed using the test proposed by Lidor et al. [41]. Scoring was awarded the same way as in the attack skills test, but the zones’ sizes differed (Figure 6). The green zone was a dimension of 1 m × 0.75 m (for 3 points), and the yellow area was 1 m × 2 m (for 2 points). The players performed this test using their standard serving technique with a self-toss. Points for the above skills were allocated by two independent observers, by the previously designated target areas, and there were no differences between the results reported by the arbiters. The competitors made 10 attempts in each test, and the sum of these ten trials was the result indicating the accuracy of the competitor. A standard ball (Molten V5M5000) was used for these tests, which players use during training and matches.

All of the above tests are designed to assess the physical performance of volleyball players and volleyball-specific skills. For this purpose, both jumping tests and running tests were used, as well as tests assessing the speed of the ball and accuracy in the performance of two elements of the volleyball game, i.e., attack and serve.

### 2.8. Side Effects and Assessment of Blinding

After the experimental trials were completed, the players answered questions related to their feelings about the possible side effects of using caffeine. Players were asked about the possible occurrence of side effects: (a) increased urine output, (b) tachycardia and heart palpitations, (c) anxiety or nervousness, (d) headache, (e) gastrointestinal problems, (f) increased vigor activeness, (g) perception of performance improvement. They were permitted to cite others issue not listed [42,43]. Additionally, the effectiveness of blinding the administered capsules was assessed. The players answered the question: “What kind of supplement do you think you took?” They could choose one of three answers: (a) “caffeine”, (b) “placebo”, and (c) “I do not know” [22].

### 2.9. Statistical Analysis

All calculations were performed using Statistica 13.3 and presented as means with standard deviations (± SD). The Shapiro–Wilk test was used in order to verify the normality of the data. The differences between the CAF vs. PLAC were identified using paired sample *t*-tests. The relative CAF–PLAC effect was calculated through effect sizes (Cohen’s *d*). Parametric effect sizes (ES) were defined as large for *d* > 0.8, moderate for *d* between 0.8 and 0.5, and as small for *d* < 0.5 [44]. Statistical significance was set at *p* < 0.05.

## 3. Results

### 3.1. Jumping Performance Assessment

The paired sample *t*-test showed no significant differences in the CM, SJ, attack jump, and block jump height between CAF and PLAC conditions (Table 1).

### 3.2. Running Performance Assessment

The paired sample *t*-test showed no significant differences between CAF and PLAC conditions in the 5 m, 10 m sprint, and agility *t*-test time (Table 1).

### 3.3. Ball Velocity during Attack and Volleyball Service

The paired sample *t*-test showed no significant differences between the CAF and the PLAC conditions in the ball velocity during the standing attack, ball velocity during the attack, and ball velocity during the serve (Table 1).

### 3.4. Volleyball Specific-Skills Assessment

The paired sample *t*-test showed significant differences between CAF and PLAC conditions in the number of points during attack accuracy test. No significant differences were observed during serve accuracy test (Table 1).

### 3.5. Side Effects and Assessment of Blinding

During the PLAC and CAF conditions, no athlete reported any side effects. In the post-exercise evaluation, 58% and 67% of respondents correctly identified the PLAC and CAF conditions, respectively.

## 4. Discussion

The purpose of this study was to investigate the acute effects of caffeine-containing chewing gum on volleyball-specific skills and physical performance in female and male volleyball players. Caffeine via caffeinated chewing gum only significantly improved the accuracy of the attack in comparison to non-caffeinated chewing gum (placebo). Caffeine-containing chewing gum did not improve other skill-based (i.e., accuracy of the serve, ball velocity during attack, and volleyball service), and physical (i.e., jumping and running performance) tests evaluated in the current study. This may suggest that caffeine in a dose of ~3 mg/kg BM via chewing gum has minor effect on physical performance and volleyball skills-based tests among volleyball players, at least with mild-moderate caffeine consumption levels.

The results of the presented study showed no significant effect of caffeine on the height of both CMJ and SJ, as well as in volleyball-specific jumping techniques. Interestingly, the vast majority of studies conducted so far have shown an improvement in the height of CMJ and SJ after caffeine supplementation in team sports including basketball [45,46], soccer [47,48,49], rugby [50], and also volleyball [15,16,17]. The reason for the lack of significant differences in the caffeine trial compared to the placebo in the presented study may be that the players were mild-moderate caffeine consumers (232 ± 170 mg of caffeine per day), not light or naive consumers as in other studies (consumed < 30 mg of caffeine per day [15]. It is worth noting that the experimental dose used in the current study was close to their normal daily caffeine intake. The previous study suggested that individuals habituated to caffeine might require a higher dose of caffeine than their habitual daily level of consumption [51,52,53,54]. This is due to the fact that chronic caffeine ingestion causes the formation of new adenosine receptors, which partially block the effects of caffeine and gradually reduce its ergogenicity during exercise [51,52,54]. However, future research should consider a sample of homogenous caffeine consumers and compare athletes with low and high habitual daily caffeine intake to confirm these suggestions.

Since jumping is considered a crucial aspect of volleyball, it is necessary to analyze the effect of caffeine on volleyball-specific movements, such as attack and block jumps [16]. The results of the current study showed no significant impact of caffeine via chewing gum on attack and block jumps. Surprisingly, the results of the current study are inconsistent with previous investigations [16,22] in which caffeine supplementation had a positive effect on those volleyball jumps. In a study by Pérez-López et al. [16], caffeine at a dose of 3 mg/kg BM via energy drink contributed to the increase in height and peak power in attack and block jumps, which were performed according to the same procedure as in the present study. Moreover, the study by Filip-Stachnik et al. [22] showed an improvement in attack jumps before and after the simulated game after supplementation of caffeinated gum at a dose of ~6 mg/kg BM. The reason for the lack of improvement in attack and block jump after caffeine supplementation observed in the current investigation might be associated with the sports level of participants. It is worth noting that the players included in the current investigation were not elite/high-performance athletes players (as they play in the 3rd national league), as recruited by the previous studies. Interestingly, some previous studies have observed that caffeine ergogenicity may vary according to training status [55,56]. Burke [57] suggested that highly trained individuals might show higher reliability in performing exercises compared to amateur athletes. Moreover, Mizuno et al. [58] found that trained men have greater adenosine A2a receptor densities than those untrained. Thus, it is possible that this increase in adenosine receptor density in trained individuals allows greater binding of caffeine to these receptors and increases the magnitude of supplementation benefits. However, those explanations seem to be speculative and more research including athletes with various training background is needed.

Volleyball players change direction frequently during training and game [59]. Therefore, in this study, we assessed speed and agility with the 5 m and 10 m sprints and the modified agility *t*-test. In addition, it was found that the ball’s velocity is significant in the context of the effectiveness of the attack and serve [60]. The results of the presented research showed that the caffeinated chewing gum did not improve the time needed to perform running tests and ball velocity, which is contrary to the results from other team sports [50,61], including volleyball [15,16]. Interestingly, those studies [15,16,50,61] were conducted with a similar dose of caffeine to that which was used in the current research (3.0 vs. 3.2 ± 0.4 mg/kg BM), but with a different method of delivery (energy drinks vs. caffeinated chewing gum). Therefore, the different pathways of caffeine absorption may be a possible explanation for the obtained results. It seems possible that caffeinated chewing gum is less effective than supplementation with caffeine in traditional forms (i.e., capsules or caffeinated drinks). The study by Sadek et al. [62] compared the pharmacokinetics of 50 mg of caffeine delivered using caffeinated chewing gum and caffeinated beverages. The result of the study by Sadek [62] showed that the average dose of caffeine released from chewing gum was about 18% lower compared to the drink. Thus, it is possible that not all of the assumed amount of caffeine is delivered. Indeed, several previous studies [21,63,64] also failed to show a significant improvement in physical performance after the intervention of caffeine via chewing gum. Future studies using caffeinated chewing gum have to include caffeine concentration analysis, especially when doses higher than 200 mg of caffeine are used [19].

In addition to evaluating the physical tests, this study aimed to assess the effect of caffeinated chewing gum on volleyball skills, which translates into points in match conditions. For this purpose, the players performed specialized tests to check the accuracy and precision of their attacks and services. To the best of the authors’ knowledge, this is the first study analyzing the effect of caffeine supplementation on skill-based volleyball performance. The results of the current study showed that caffeinated chewing gum improved the number of points obtained during the accuracy attack test. Although the results during the accuracy service test did not reach a significant improvement, a positive trend (ES = 0.57) in caffeine condition has been observed. It is worth noting that the accuracy assessed in the current study may demand maintaining vigilance, alertness, and related parameters. It has been shown that low doses of caffeine (i.e., ~200 mg or ~3 mg/kg BM as used in the current study) improve vigilance, alertness, mood, and cognitive processes during exercise, but do not alter the peripheral whole-body responses to exercise [12]. Therefore, it is possible that the ergogenic effect of low caffeine observed in accuracy tests may appear to result from alterations in the central nervous system. However, from a practical view point, the effects of caffeine supplementation seem to be variable [12], and athletes need to determine whether the ingestion of low caffeine doses is ergogenic on an individual basis.

The results of the current study showed no side effects when caffeinated chewing gum was used. Thus, it confirmed that using caffeine via chewing gum reduces the risk of possible side effects compared to other forms of this supplement [16,65]. However, the absence of side effects may also be caused by the habituation of caffeine by participants. It has been shown that individuals who consume caffeine on a regular basis, may develop a tolerance to it [66]. Further, caffeine users typically experience caffeine withdrawal syndrome, resulting in headaches, anxiety, and nervousness [67]. Interestingly, those side effects were not observed during the placebo conditions. Thus, it can be suggested that mild to moderate caffeine users can refrain from consuming caffeine and perform physical activities without a decrease in performance and common withdrawal symptoms. This approach may be an innovative issue in the context of caffeine supplementation by athletes, but further research is needed in this direction.

Beyond the present study’s strengths, its limitations should also be considered. The current study did not include any biochemical analyzes. Only one caffeine dose of caffeine (i.e., ~3 mg/kg BM) was assessed; thus it is not known whether higher doses would improve analyzed outcomes. In addition, the experimental dose of caffeine was similar to the daily caffeine intake of the athletes, and it is possible that higher caffeine doses may provide performance benefits. Future studies should be conducted with a larger number of participants and involve a comparable number of men and women. Lastly, it is also worth considering to reduce the number of tests in future studies to exclude possible fatigue effects during following performance tests.

## 5. Conclusions

The results of the present study indicated that the acute intake of caffeine (~3 mg/kg BM) via chewing gum significantly improves the results of attack accuracy in volleyball players. However, this supplement did not improve the results in jumping tests, running tests, and other volleyball specific-skills, including speed tests and accuracy during the volleyball, serve. Additionally, caffeine supplemented in such a dose and form minimizes the risk of side effects. From a practical viewpoint, the benefits of caffeinated chewing gum seem to be minor in comparison to caffeine capsules or energy drinks, at least in athletes habituated to caffeine consumption.

## Figures and Tables

**Figure 1 nutrients-15-00091-f001:**
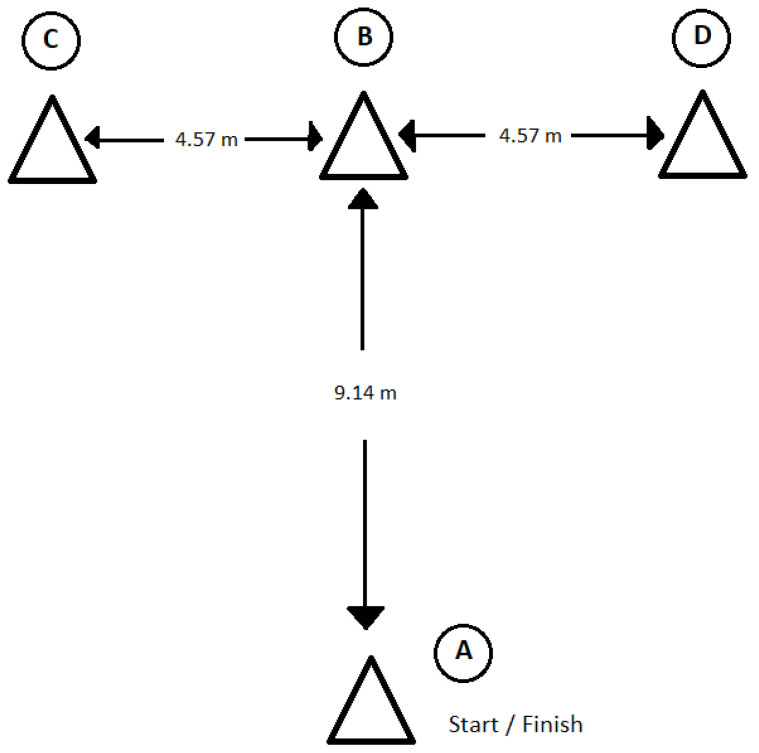
The course of modified agility *t*-test.

**Figure 2 nutrients-15-00091-f002:**
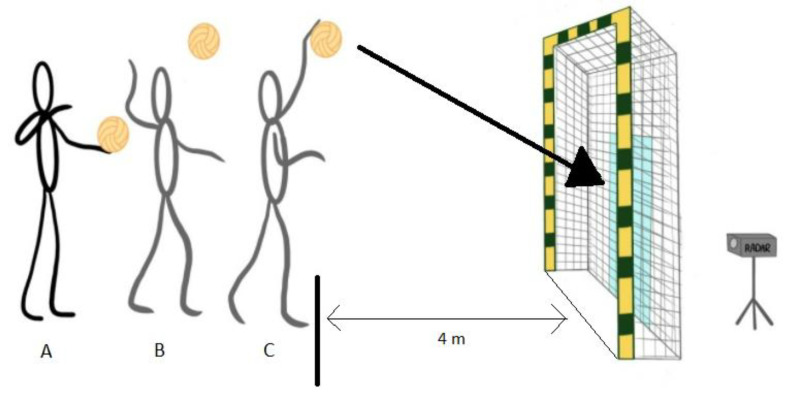
The maximal standing spike test.

**Figure 3 nutrients-15-00091-f003:**
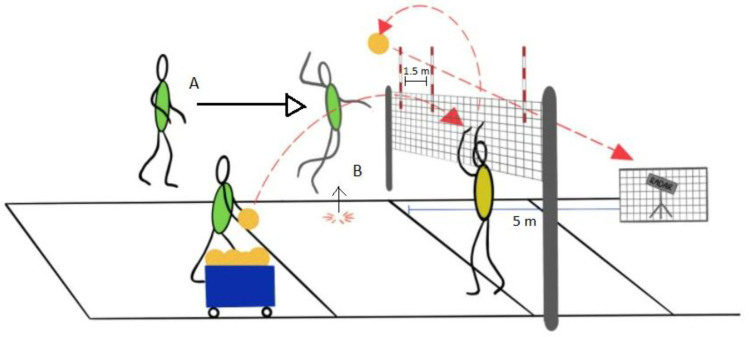
The attack test.

**Figure 4 nutrients-15-00091-f004:**
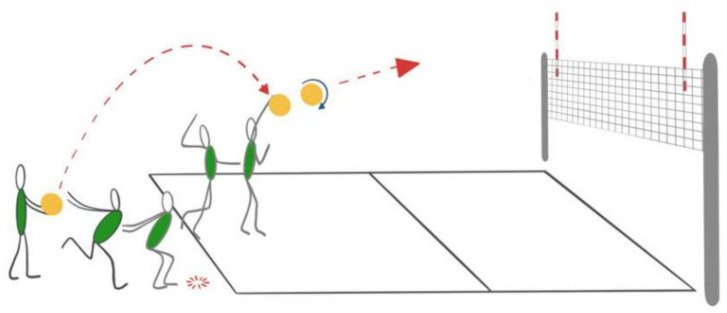
The volleyball service test.

**Figure 5 nutrients-15-00091-f005:**
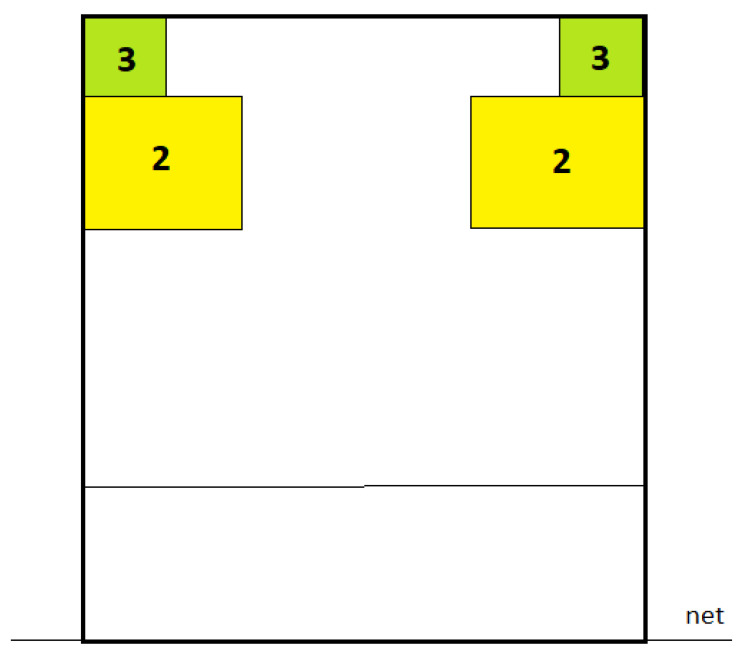
Division of the court (9 m × 9 m) into point zones to assess the accuracy of the attack.

**Figure 6 nutrients-15-00091-f006:**
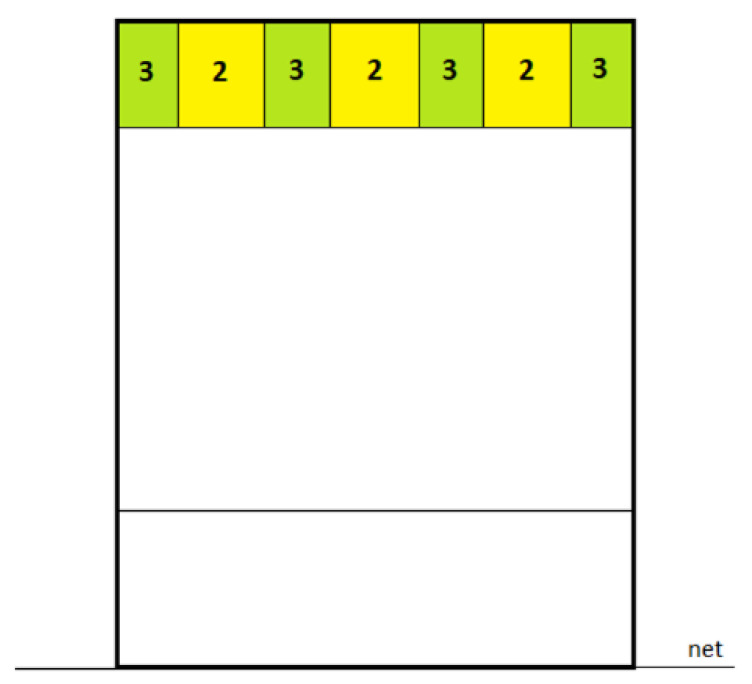
Division of the court (9 m × 9 m) into point zones to assess the accuracy of the volleyball service.

**Table 1 nutrients-15-00091-t001:** Comparison of the caffeinated chewing gum (~3.2 mg/kg of caffeine) and non-caffeinated chewing gum (placebo) trial in volleyball-specific skills and physical performance tests.

Variable	CAF (95% CI)	PLAC (95% CI)	*p*	*d* (95% CI)
CMJ (cm)	51.2 ± 11.2 (44.09–58.28)	51.0 ± 11.4 (43.77–58.25)	0.820	0.02 (−0.78–0.82)
SJ (cm)	39.1 ± 7.8 (34.11–44.09)	40.9 ± 9.6 (34.75–46.95)	0.230	0.21 (−0.60–1.00)
Attack jump (cm)	61.4 ± 14.9 (51.95–70.86)	62.4 ± 13.9 (53.61–71.25)	0.342	0.07 (−0.73–0.87)
Block jump (cm)	48.4 ± 10.6 (41.68–55.09)	48.4 ± 11.6 (41.02–55.73)	0.995	0.00 (−0.80–0.80)
5 m sprint (s)	0.95 ± 0.11 (0.89–1.02)	0.95 ± 0.11 (0.88–1.03)	1.000	0.00 (−0.80–0.80)
10 m sprint (s)	1.69 ± 0.12 (1.61–1.76)	1.68 ± 0.13 (1.60–1.76)	0.619	0.08 (−0.72–0.88)
Agility *t*-test (s)	9.44 ± 0.69 (9.01–9.88)	9.45 ± 0.77 (8.96–9.94)	0.952	0.01 (−0.79–0.81)
Standing attack (km/h)	82 ± 11 (75–89)	79 ± 12 (71–86)	0.274	0.26 (−0.55–1.05)
Attack (km/h)	85 ± 14 (76–94)	81 ± 13 (73–89)	0.119	0.30 (−0.52–1.09)
Serve (km/h)	88 ± 14 (79–97)	86 ± 13 (78–95)	0.254	0.15 (−0.66–0.94)
Attack accuracy (points)	18 ± 3 (16–19)	15 ± 4 (13–18)	0.023 *	0.85 (−0.01–1.65)
Serve accuracy (points)	12 ± 4 (9–15)	10 ± 3 (8–13)	0.140	0.57 (−0.27–1.36)

CAF: caffeine; PLAC: placebo; CMJ: countermovement jump; SJ: squat jump; *d*: Cohen’s *d* effect size; CI: Confidence Interval. * Statistically significant difference (*p* < 0.05) compared to the placebo. All data are presented as mean ± standard deviation.

## Data Availability

The data presented in this study are available on request from the corresponding author.

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
