# Peer review of "The Effect of Caffeinated Chewing Gum on Volleyball-Specific Skills and Physical Performance in Volleyball Players"

_nutrients, 2022, doi:10.3390/nu15010091_

Round 1
Reviewer 1 Report
The paper is well-written and easy to read. I would recommend some revisions. See below.
The study examines caffeine consumption via chewing gum on volley-specific performance and skills. This study looks to expand on previous research, showing caffeine supplements may positively affect volleyball performance. The novel aspect of the study is the use of caffeinated gum that can be easily administered immediately for and during play.
The study suggests caffeinated chewing gum does not improve physical or skill-based volleyball performance. The subjects in the study were habitual caffeine users that were asked not to consume caffeine 24 hours before testing. A shortcoming of the research methodology is that experimental caffeine consumption was similar to the participant's daily caffeine consumption. Previous literature looking at performance and caffeine consumption suggests that subjects require a higher dose of caffeine than their usual daily intake to see positive effects on athletic performance. I feel the authors address this issue towards the end of the discussion, but should also stress earlier in the paper that the subjects were habitual caffeine users and they received a similar experimental dose to their normal daily intake. The authors state this point very well in the last sentence of the first paragraph in the discussion lines 268 - 270.
The authors could also suggest that the placebo group did not significantly decrease athletic performance. This indicates that habitual caffeine users may refrain from caffeine consumption and perform volleyball-related skills without a decrease in performance.
A strong point of the paper is the authors do an admiral job of describing and administering volleyball-specific skills tests that can be repeated in future research. Furthermore, graphs and diagrams are easy to read and help explain their methods.
The authors bring up a couple of points in the discussion that I feel they could elaborate. The first is why they feel the current study did not show improvements in the height of CMJ and SJ after caffeine supplementation, as shown in other research involving team sports. Lines273-275.
Further on in the discussion, the authors suggest a reason their subjects did not improve the height of specialized jumps after caffeine supplementation may be that their subjects were not elite volleyball players. Lines 296-297. However, they do not suggest why caffeine consumption significantly affects elite athletes more than sub-elite athletes.
Lastly, in lines 307 to 309 the authors state that caffeinated chewing gum did not show similar positive effects on performance as caffeine consumption by energy drinks. Sadex et al. (Journal of Caffeine Research.Dec 2017.125-132.) showed no significant differences between caffeine delivery via chewing gum and beverage consumption. Authors should suggest why they feel the caffeine delivery method in their study would have a different result than in previous findings.
Overall the study provides valuable insight into performance testing and volleyball-specific skills tests. The authors should refine their discussion to focus more on the caffeine dosage than the drug's delivery method unless they have a reasonable explanation for why their findings differ from previous research.
Author Response
We sincerely thank the Reviewers and the Editors of Nutrients Journal for their careful peer-reviewing of the manuscript and for their valuable comments provided in the review letter. Below, we have included a response letter where we have replied item-by-item to the comments provided by the Reviewers. We have addressed your specific comments in the manuscript using the *Highlight* function in Microsoft Word. We feel that the manuscript is improved in light of the suggested changes.

Reviewer 2 Report
The aim of this paper, according to the authors, is to investigate the effect of caffeinated chewing gum on volleyball-specific skills and physical performance in volleyball players. The topic is interesting and the paper is well written. Still, it has some weaknesses that should be addressed.
Introduction
Interesting information would be missing in the introduction.
It would be interesting to expand the introduction, adding information on how caffeine consumption is currently regulated in sports competition.
It is necessary to update the bibliographic references in some sections, e.g. in the sentence of line 33 “Caffeine is considered the most popular psychoactive drug...”, and line 35 “it is known that caffeine has a positive effect on various types of exercise...”.
Lines 33 to 62. It would be appropriate to add information on current caffeine consumption in athletes. Authors should explain if there are limits to the amount of caffeine athletes can consume in sport competition.
Materials and Methods
The sample is too small, making it difficult to obtain robust results. Furthermore, it would have been interesting to record other variables of interest.
Line 75 to 85. Authors must justify the inclusion criteria, especially why include the same phase of the menstrual cycle in women and why not be positive smokers.
Line 96. Please clarify the criteria for classification into mild and moderate caffeine users.
Lines 125-215. Authors should make an effort to summarize the explanations of the practical tests used to assess volleyball-specific skills and physical fitness.
Line 218. In connection with to checking player feelings and assessment of blinding, are there one or more questions? If there are more than one, please rate the type of questions that were asked.
Results
All the results they might be displayed in a single table.
It is recommended to use "d" instead of "ES" in all tables.
It is necessary to add in tables 1 to 4 a new column with the p-value with three decimal places (T-test). Along with the value of d, the authors can add the value of 95% CI that improves the results.
In the table title, authors may consider adding "Comparison of..." or "Adjusted means and effect sizes for...".
It is recommended to remove "CAF vs. PLAC" in all column headers -It is obvious-, and in the footer to add d: Cohen's d effect size.
That will involve restructuring the text of the results.
Discussion and conclusions
They are adjusted to the results.
References
It would be appropriate to remove the month of publication in the references.
Author Response

(The authors gave the same response as above.)

Round 2
Reviewer 2 Report
The authors have now submitted a revised version providing the changes suggested in my review. The topic is interesting and novel, however, the research design could not be appropriate for the editor because the research sample could be considered unacceptable for a high impact journal.
Author Response
We sincerely thank the Reviewers and the Editors of Nutrients Journal for their careful peer-reviewing of the manuscript and for their valuable comments. Nevertheless, we believe that the manuscript after applying the suggested changes significantly gained in value.